# Comprehensive discovery of CRISPR-targeted terminally redundant sequences in the human gut metagenome: Viruses, plasmids, and more

Ryota Sugimoto[1], Luca Nishimura[1,2], Phuong Thanh Nguyen[1,2], Jumpei Ito[3], Nicholas F. Parrish[4], Hiroshi Mori[5], Ken Kurokawa[6], Hirofumi Nakaoka[7], Ituro Inoue[1] *

1 Human Genetics Laboratory, National Institute of Genetics, Research Organization of Information and Systems, Mishima, Shizuoka, Japan, 2 The Graduate University for Advanced Studies, SOKENDAI, Mishima, Shizuoka, Japan, 3 Division of Systems Virology, Department of Infectious Disease Control, International Research Center for Infectious Diseases, Institute of Medical Science, The University of Tokyo, Minato-ku, Tokyo, Japan, 4 Genome Immunobiology RIKEN Hakubi Research Team, Center for Integrative Medical Sciences, RIKEN, Tsurumi-ku, Yokohama, Kanagawa, Japan, 5 Genome Diversity Laboratory, National Institute of Genetics, Research Organization of Information and Systems, Mishima, Shizuoka, Japan, 6 Genome Evolution Laboratory, National Institute of Genetics, Research Organization of Information and Systems, Mishima, Shizuoka, Japan, 7 Department of Cancer Genome Research, Sasaki Institute, Chiyoda-ku, Tokyo, Japan

* itinoue@nig.ac.jp

**Data Availability Statement:** All relevant data are within the manuscript and its Supporting Information file are available in the Zenodo

## Abstract

Viruses are the most numerous biological entity, existing in all environments and infecting all cellular organisms. Compared with cellular life, the evolution and origin of viruses are poorly understood; viruses are enormously diverse, and most lack sequence similarity to cellular genes. To uncover viral sequences without relying on either reference viral sequences from databases or marker genes that characterize specific viral taxa, we developed an analysis pipeline for virus inference based on clustered regularly interspaced short palindromic repeats (CRISPR). CRISPR is a prokaryotic nucleic acid restriction system that stores the memory of previous exposure. Our protocol can infer CRISPR-targeted sequences, including viruses, plasmids, and previously uncharacterized elements, and predict their hosts using unassembled short-read metagenomic sequencing data. By analyzing human gut metagenomic data, we extracted 11,391 terminally redundant CRISPR-targeted sequences, which are likely complete circular genomes. The sequences included 2,154 tailed-phage genomes, together with 257 complete crAssphage genomes, 11 genomes larger than 200 kilobases, 766 genomes of *Microviridae* species, 56 genomes of *Inoviridae* species, and 95 previously uncharacterized circular small genomes that have no reliably predicted protein-coding gene. We predicted the host(s) of approximately 70% of the discovered genomes at the taxonomic level of phylum by linking protospacers to taxonomically assigned CRISPR direct repeats. These results demonstrate that our protocol is efficient for *de novo* inference of CRISPR-targeted sequences and their host prediction.

repository: https://doi.org/10.5281/zenodo.5500088 The source codes used in this study are available in the Zenodo repository: https://doi.org/10.5281/zenodo.5500079.

**Funding:** R.S. is funded by the Research Organization of Information and Science (https://www.rois.ac.jp). L.N. is funded by JSPS Research Fellow DC1 21J22509 (https://www.jsps.go.jp/english). P.T.N is funded by MEXT Doctoral Scholarship 180243 (https://www.mext.go.jp). J.I. is funded by JSPS KAKENHI JP20K15767 (https://kaken.nii.ac.jp/grant/KAKENHI-PROJECT-20K15767/) and JSPS Research Fellow PD 19J01713 (https://www.jsps.go.jp/english). I.I. is funded by JSPS KAKENHI 20K21405 (https://kaken.nii.ac.jp/grant/KAKENHI-PROJECT-20K21405/). The other authors received no specific funding for this work. The funders had no role in study design, data collection and analysis, decision to publish, or preparation of the manuscript.

**Competing interests:** The authors have declared that no competing interests exist.

## Author summary

The evolution and origins of viruses are long-standing questions in the field of biology. Viral genomes provide fundamental information to infer the evolution and origin of viruses. However, viruses are extraordinarily diverse, and there are no single genes shared across entire species. Several methods were developed to collect viral genomes from metagenome. To infer viral genomes from metagenome, previous approaches relied on reference viral genomes. We thought that such reference-based methods may not be sufficient to uncover diverse viral genomes; therefore, we developed a pipeline that utilizes CRISPR, a prokaryotic adaptive immunological memory. Using this pipeline, we discovered more than 10,000 positively complete CRISPR-targeted genomes from human gut metagenome datasets. A substantial portion of the discovered genomes encoded various types of capsid proteins, supporting the contention that these sequences are viral. Although the majority of these capsid-protein-coding sequences were previously characterized, we notably discovered *Inoviridae* genomes that were previously difficult to infer as being viral. Furthermore, some of the remaining unclassified sequences without a detectable capsid-protein-encoding gene had a notably low protein-coding ratio. Overall, our pipeline successfully discovered viruses and previously uncharacterized presumably mobile genetic elements targeted by CRISPR.

This is a *PLOS Computational Biology* Methods paper.

## Introduction

Viruses are not counted among the domains of life, but they are the most numerous biological entity, presumably containing the most genetic material on Earth. The number of viruses infecting microbial populations is estimated at $10^{31}$ [1]. Viruses exist in all environments and infect all cellular organisms. In the historical experiments of Hershey and Chase [2], a virus was first used to show that DNA is the genetic material, and the first complete genome sequence was viral [3,4]; however, despite these historical efforts, the evolution and origins of viruses are poorly understood, because of the nature of viruses. First, viral genomes are incredibly diverse in size (ranging from several thousand to a few million bases) and genomic content (encoding various nonoverlapping sets of genes). In fact, there are no genes that are universally present among all viruses, indicating that a monophyletic explanation of viral origin is implausible [5,6]. Second, most viral genes show little to no homology to cellular genes. Viral genomes contain sparse sequences from which to infer shared lineage or evolutionary history with cellular life. Considering these confounding biological features, we were motivated to uncover viral genomic sequences without relying on either marker genes or known sequences.

Metagenomics [7] has become a popular method to sequence and detect viral genomes. Metagenomics sequences all genetic materials extracted from a given environmental sample. The advantage of a metagenomic approach is that, when using a sufficient depth of sequencing, presumably all genetic information of the viruses in the sample can be obtained. However, because metagenomic approaches obtain a mixture of cellular and viral genomic information, it is necessary to distinguish viral from cellular sequences.

Several methods have been developed to extract viral sequences from metagenomic data after assembling sequences into those derived from presumably contiguous DNA molecules called contigs. One virus-detection software, VirSorter [8], detects regions within contigs

enriched with viral "hallmark genes," viral-like genes, uncharacterized genes, and shortened genes. This program uses a custom-made protein database to detect viral proteins, including viral reference proteins and predicted proteins from virome samples. Another virus-detection method, VirFinder [9], is a k-mer–based machine-learning program that uses a logistic regression model trained by RefSeq viral and prokaryotic genomes. It uses the k-mer features of a given contig as input and outputs the probability that the contig is a virus. Although these methods have been used to uncover viral diversity from metagenomics data [10], both rely on similarity to known viral genomes. Therefore, as reference-based approaches, they cannot uncover viral lineages that are extremely distant from, or unrelated to, known viral genomes, if such lineages exist. Thus, we sought to develop a nonreference-based analytical pipeline to detect viral sequences from metagenomes; however, we were tasked with the question "What information could be used for this purpose?" The underlying biology of the clustered regularly interspaced short palindromic repeats (CRISPR) system, a prokaryotic form of adaptive immunological memory [11], provides a potential resource in this context. After viral infection or horizontal plasmid transfer, some archaeal and bacterial cells incorporate fragments of "nonself" genetic materials in specialized genomic loci between CRISPR direct repeats (DRs). The incorporated sequences, called "spacers," are identical to part of the previously infecting mobile genetic element. Thus, the genetic information encoded in CRISPR spacers can be inferred as likely viral and distinguishable from the genetic material of the organism encoding CRISPR, which is most often cellular, but potentially also viral [12,13].

CRISPR spacers have been used to detect viral genomes [14–18] and predict viral hosts [19,20]. They have previously been extracted from assembled bacterial genomes to assess CRISPR "dark matter", revealing that 80%–90% of identifiable material matches known viral genomes [17,18]. In the current study, we extended this conceptual approach to the enormous amount of unassembled short-read metagenomic data. CRISPR repeats are relatively easily identifiable, particularly compared with unknown viral sequences. This trait allows the search of massive metagenomic datasets for reads comprised in part as CRISPR DR sequences; in turn, unknown sequences inferred as CRISPR spacers can be extracted directly from the raw reads [21,22]. By analyzing these reads and the contigs assembled from them, we successfully extracted 11,391 terminally redundant (TR) CRISPR-targeted sequences ranging from 894 to 292,414 bases. These sequences are expected to be complete or near-complete circular genomes that can be linked to their CRISPR-targeting hosts. The discovered sequences include 2,154 tailed-phage genomes, together with 257 complete crAssphage genomes [23,24], 11 genomes larger than 200 kilobases (kb), 766 *Microviridae* genomes, 56 *Inoviridae* genomes, 5,658 plasmid-like genomes, and 2,757 uncharacterized genomes. Although the majority of the discovered sequences that were larger than 20 kb were mostly characterized viral or plasmid genomes, a substantial portion of sequences smaller than 20 kb was not recorded in either plasmid or viral databases. Furthermore, some previously uncharacterized small genomes had notably low coding ratios, which indicate that these elements might have unknown noncoding genetic features. These results demonstrate that our pipeline can discover CRISPR-targeted mobile genetic elements (MGEs) either previously characterized or uncharacterized.

## Results

### Extraction of CRISPR-targeted sequences

We analyzed human gut metagenomes, as they serve as an "ecosystem" with the most abundant metagenomic data available. We downloaded 11,817 human gut metagenome datasets equivalent to 50.7 Tb from the European Nucleotide Archive FTP server. FASTQ files were preprocessed and assembled to 180,068,349 contigs comprising 767.7 Gb of data (S1 Table).

We discovered 11,223 unique CRISPR DRs from the assembled contigs that were used to extract CRISPR spacers from raw reads, resulting in 1,969,721 unique CRISPR spacers (S1 Fig and see here: https://doi.org/10.5281/zenodo.5500088). These spacers were then used as queries to identify candidate protospacers (i.e., contigs containing the spacer sequence, not within a CRISPR locus). Spacers were mapped to CRISPR masked contigs using a minimum sequence identity threshold of 93%. We chose this identity threshold to capture the escaped mutants, i.e., viruses that escaped CRISPR targeting by introducing mutations to the protospacer loci. To increase specificity, we verified that the 5′- and 3′-adjacent sequences of spacer-mapped positions were not similar to each other or the spacer-associated DR. A total of 164,590,387 candidate protospacer loci, attributed to 1,114,947 unique spacers (56.6% of all unique spacers), were identified (see here https://doi.org/10.5281/zenodo.5500088). This is a substantially higher discovery rate than that reported previously (~7% [17]) in a study that used National Center for Biotechnology Information (NCBI) nucleotide sequences for protospacer discovery. Although the genuine protospacers from a viral genome are expected to be colocalized in a relatively small region, the false-positive protospacers are expected to be scattered across the metagenome contigs randomly. To further reduce the false-positive hits, spacers were clustered based on protospacer co-occurrence and used to extract contigs targeted by more than 30% of members of a spacer cluster (S2 Fig). This process effectively removes false-positive protospacers that are randomly distributed across the assembled contigs. Finally, 764,883 gapless CRISPR-targeted sequences (15.9 Gb) were extracted. Among them, 11,391 unique sequences were identified as TR [25]; we expected that they were initially complete or near-complete circular MGEs. The size of the CRISPR-targeted TR sequences ranged from 894 to 292,414 bases (S2 Table and see here: https://doi.org/10.5281/zenodo.5500088).

We then investigated protein-coding genes encoded in CRISPR-targeted sequences and identified 240,369 protein-coding genes among all unique TR sequences. Protein sequences were clustered based on a 30% sequence identity threshold, resulting in 31,204 clusters. Each representative sequence was used as a query for three jackhmmer iterations, to build Hidden Markov models (HMM), which were then used to search the Protein Data Bank (PDB) [26]. Finally, 10,641 representative sequences, including 110,386 predicted protein sequences, were annotated (HHsearch probability > 80 and E-value < 1e–3) (See here: https://doi.org/10.5281/zenodo.5500088).

## Classification of TR sequences

The evaluation of TR sequence length revealed a multimodal distribution with a distinct trough at 20 kb (Fig 1A). For reference, we termed the 8837 TR sequences that were shorter than 20 kb as "small" and the 2554 TR sequences that were longer than 20 kb as "large." This simple classification was previously used to infer capsid morphology [27,28]. Among the large TR sequences, 2047 (80.1%) encoded HK97 fold capsid proteins, a definitive gene of *Duplodnaviria* [29]. Phage portal proteins were encoded by 2163 large TR sequences (84.7%), indicating that most large TR sequences are from *Caudovirales*, also known as tailed phages (Fig 1B). Among the small TR sequences, 766 (8.7%) encoded *Microviridae* major capsid proteins (MCPs) [30], and 56 (0.6%) encoded *Inoviridae* major coat proteins. We propose that this portion of small TR sequences are likely viruses with a non-tailed morphology (Fig 1B) [31,32]. Finally, 107 (1.2%) small TR sequences encoded HK97 fold capsid proteins. We also sought to identify any TR sequences encoding vertical jelly roll fold (vJR) capsids, a definitive gene of *Varidnaviria* [33]; however, we failed to find a significant hit using our search criteria.

We considered that a fraction of the TR sequences that lacked detectable capsid genes included plasmids. Therefore, we examined whether these TR sequences encode proteins that

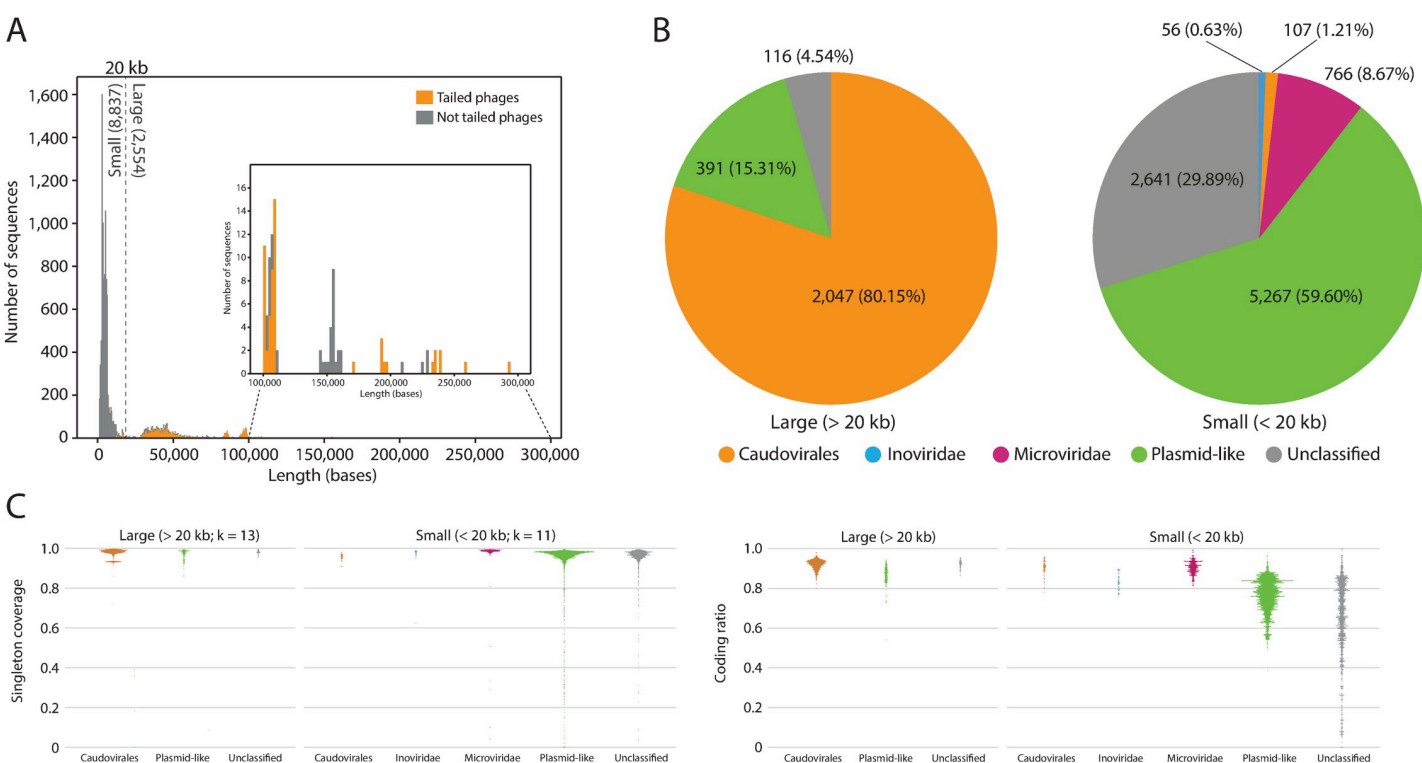

**Fig 1. Classification and genetic features of CRISPR-targeted TR sequences. (A)** Length distribution of TR sequences. We used the HK97 capsid and portal proteins as tailed-phage signature genes. The dotted line at 20 kb represents an arbitrary cut-off between small and large sequences. Sequences longer than 100 kb are shown in the inset. **(B)** Results of the classification of TR sequences. Sequences encoding a detectable capsid gene were classified to a viral taxon according to capsid type, as follows. *Caudovirales*: HK97 fold capsid; *Inoviridae*: *Inoviridae* MCP; and *Microviridae*: *Microviridae* MCP. The capsid-less TR sequences with ParA, ParB, ParM, and/or MoBM were classified as Plasmid-like. The remaining sequences were labeled as "Unclassified." **(C)** Distribution of singleton coverage and coding ratio. Selected k-values were higher in large TR sequences, to avoid doubletons by chance.

are characteristic of plasmids. We identified 386 large TR sequences (15.1%) and 957 small TR sequences (10.8%) encoding plasmid partitioning proteins A, B, or M. Furthermore, 187 large (7.3%) and 4554 small (51.5%) TR sequences encoded MoBM relaxase, a protein that is required for initiating conjugation. Thus, 391 large (15.3%) and 5267 small (59.6%) TR sequences are likely plasmids or have life cycles similar to that of plasmids.

To scrutinize other genomic features, such as repeats and noncoding regions, the k-mer singleton coverage and coding ratio for each classified and unclassified TR sequence were investigated (Fig 1C). Singleton coverage is the number of k-mer singletons from a given contig divided by its length; the value approaches 1 if the sequence does not contain repeats. For the large TR sequences, both classified and unclassified sequences had singleton coverages and coding ratios close to 1, indicating that these sequences are densely occupied by protein-coding genes and have few repeats. Conversely, plasmid-like and unclassified small TR sequences had a wider distribution of singleton coverages and coding ratios, indicating that some of these sequences may have a large proportion of noncoding regions and repeats.

## Predicted CRISPR-targeting hosts of TR sequences

As our approach uses CRISPR spacers to extract CRISPR-targeted sequences, we hypothesized that the relationship between a virus and the targeted host could be resolved for the majority of TR sequences. CRISPR DR sequences were searched on RefSeq genomes and taxonomically assigned. CRISPR DRs are shared between distant species through horizontal gene transfer

(HGT) [34]. To prevent misassignment of targeting host because of shared DRs, we did not taxonomically assign DRs that were shared between different lineages in a given taxonomic level, and those DRs did not contribute to the targeting host prediction. Based on counts of protospacers associated with taxonomically assigned DRs, 7937 TR sequences (69.7%) were resolved to a targeting host at the phylum level (Figs 2A and S3 and S2 Table), and 6083 TR sequences (53.4%) were resolved to a targeting host at the order level (S4 Fig and S2 Table). The most frequent host was *Firmicutes*, followed by *Bacteroidetes* and *Actinobacteria*. Notably, these are the most common bacteria in the human intestine [35]. In addition, 1418 TR sequences (12.5%) had putative host ambiguity between multiple phyla. Although some of these TR sequences were associated exclusively with monoderm or diderm phyla, there was exceptional targeting host ambiguity between *Firmicutes* and *Verrucomicrobia*, which crosses the monoderm–diderm boundary (S5 Fig).

### *Firmicutes*–*Verrucomicrobia* multiple infecting viruses are suspicious

These curious results prompted us to verify whether these sequences are indeed targeted to both *Firmicutes* and *Verrucomicrobia*. We focused on one of these sequences, dubbed amb-1 (note in S2 Table), which is 54,793 bases long and encodes an HK97 fold capsid and portal protein genes, suggesting it is likely a tailed phage. We discovered 676 protospacers in amb-1, 194 of which were associated with the DR sequence "CGTCGCACTCCGCAAGGAGTGCGTG GATTGAAAC" (DR1), whereas the remaining 422 were associated with the DR sequence "GTCGCTCTCCGCAAGGAGAGCGTGGATAGAAATG" (DR2). DR1 and DR2 pairwise alignment yielded only four mismatches (82.9% identity). Our criteria define this level of sequence identity as sufficiently low for separate taxonomic assignments. DR1 aligned perfectly to the *Clostridia* genomes NZ_PSQF01000044 and NZ_NFID01000002.1, and DR2 aligned perfectly to *Akkermansia muciniphila* genomes. The DR sequence similarity could be explained by HGT of the CRISPR–Cas system between these species. In fact, we found a phage portal and tail gene from the adjacent region of the DR2-aligned positions in CP027011.1, indicating that the CRISPR–Cas system associated with DR2 might have been recently introduced to the *Akkermansia muciniphila* genome by phage integration. The possibility of HGT limits the utility of the DR–host connection for inferring the actual targeting host of amb-1. We searched for signs of HGT between amb-1 and its genuine host, to complement the DR–host connection-based method. We used amb-1 to query the nr database and found that it partially aligned with the *Oscillibacter* genome AP023420.1 (query coverage of 6%, with 67.41% identity), but no significant hit against *Akkermansia muciniphila* was found. Thus, there is no substantial evidence to suggest that amb-1 is being targeted by multiple phyla. Although we focused on one TR sequence in this validation, the signs of phage integration and horizontal transfer of CRISPR–Cas between *Clostridia* and *Akkermansia muciniphila* suggest that the host–DR connection can not be used to infer genuine targeting hosts, particularly between these species.

### Predicted targeting hosts above the taxonomic level of order are consistent

As we showed on an *ad hoc* basis, the CRISPR–Cas system undergoes frequent HGT between species. Because of unrecorded HGTs that could have occurred very recently, the DR-to-RefSeq alignment method might cause misinterpretations in the prediction of targeting hosts. Therefore, the TR sequences assigned to a single targeting host taxon still require further assessment. To complement DR-to-RefSeq–based host prediction, we used the tRNA genes [36] encoded in TR sequences. We found that 552 TR sequences encoded a total 1124 tRNA genes. These tRNA gene sequences were searched for RefSeq bacterial genomes (95%

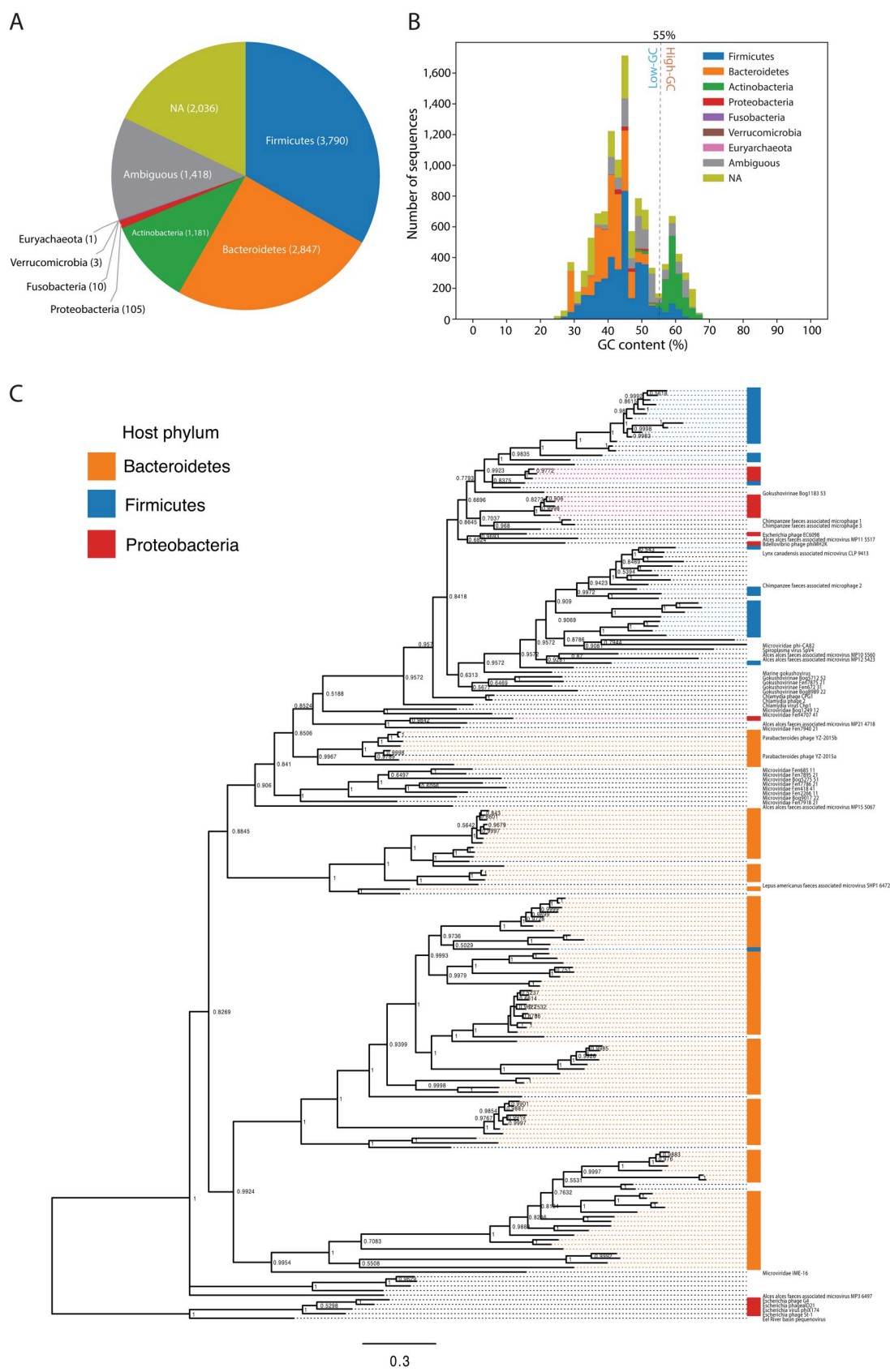

**Fig 2. Predicted targeting hosts of CRISPR-targeted TR sequences. (A)** The targeting host composition of TR sequences. Hosts were predicted by mapping CRISPR DR sequences to the RefSeq database. Sequences containing ≥10 protospacer loci but less than 90% associated DR taxa exclusiveness were classified as an ambiguous targeting host. When ≥10 protospacers could not be assigned to a taxon, the predicted targeting host was denoted as not available (NA). **(B)** Predicted targeting host distribution according to GC content. The dotted line indicates the low- and high-GC content boundary, at 55%. **(C)** Bayesian phylogeny of *Microviridae* major capsid proteins. A total of 159 representative major capsid protein sequences from this study, and 43 RefSeq sequences were used for analysis. Taxa without a name denote the *Microviridae* species from this study, and taxa with text denote *Microviridae* species from RefSeq. Taxa were annotated based on predicted targeting hosts. The phi X174 clade was selected as the outgroup.

minimum query coverage, with 95% minimum sequence identity); 288 tRNA gene sequences were aligned to 82 bacterial species genomes, connecting 97 TR sequences to potential hosts (see here: https://doi.org/10.5281/zenodo.5500088) The comparison of the tRNA-based predicted hosts with the DR-to-RefSeq–based predicted targeting hosts revealed a 93% agreement at the taxonomic level of order, and more at higher taxonomic levels (S6A Fig). The predicted host agreement dropped to 75% at the genus taxonomic level, suggesting that the DR-to-RefSeq–based method is less reliable for taxonomic levels lower than order.

During the review of this manuscript, a study of an enormous gut virome was published [37]. The authors of that study developed the Metagenomic Gut Virome (MGV) catalog, in which the viral genomes were assigned to predicted hosts based on CRISPR spacer alignment. Unlike our method, their CRISPR spacers were extracted from the taxonomically assigned metagenomic contigs recorded in the Unified Human Gastrointestinal Genome database. Thus, we considered that their predicted hosts could be used to assess our results. We found that 2180 TR sequences from our study were also recorded in MGV (see here: https://doi.org/10.5281/zenodo.5500088). For this sequence comparison, we used stricter criteria (85% query coverage, with 95% minimum sequence identity), to avoid host ambiguity between the relatively distant viral species. The predicted hosts were compared for each shared sequence between the two studies, and we found 95% agreement at the taxonomic level of order, and more at higher taxonomic levels (S6B Fig).

### *Actinobacteria* is the corresponding targeting host of high-GC-content TR sequences

We calculated the GC content [38] of the TR sequences, to determine whether these correspond to the GC content of the targeting host predicted by the DR–host connection-based method (Fig 2B). Among the 2050 high-GC (GC% >55%) TR sequences, 1109 (54%) and 249 (12.1%) were predicted to be targeted by *Actinobacteria* and *Firmicutes*, respectively, and the targeting host was undetermined for 222 TR sequences (10.8%). The large fraction of high-GC TR sequences that were predicted to be targeted by *Actinobacteria* likely indicate the genomic adaption of parasitic genetic elements that infect and routinely become targeted by the CRISPR systems of high-GC Gram-positive bacteria (e.g., *Actinobacteria*).

### *Microviridae* species encountered a cross-phylum host-switching event

Host range within a viral lineage was further assessed by phylogenetic analysis of the TR sequences that were determined to represent *Microviridae* species. The predicted targeting hosts of putative *Microviridae* species from our study were *Bacteroides*, *Firmicutes*, and *Proteobacteria* (S2 Table), indicating a broad host range for *Microviridae* species. The molecular phylogeny of the *Microviridae* major capsid protein clearly segregated sequences based on their targeting host (Fig 2C). Interestingly, most of the known *Escherichia coli*-infecting viral species (such as phiX174) and other *Proteobacteria*-infecting species (such as phi MH2K [39]) that were used as a reference in this phylogeny were split into two clades, and the clade containing

the latter was within a clade of TR sequences targeted by *Firmicutes*. The nested phylogenetic structure of capsids encoded by TR sequences that were predicted to represent *Microviridae* species may indicate that host switching, a critically important topic in viral evolution [40,41], occurred within the viral lineage.

## CRISPR-targeted noncoding elements

Intrigued by the lower coding ratio detected in some unclassified small TR sequences, we selected two representative sequences with a notably low coding ratio (<0.3) for further manual inspection. These two sequences had distinguishable sequence similarity and GC content. For simplicity, we dubbed them circ-1 and circ-2 (note in S2 Table). Circ-1 represented 84 small TR sequences (see here: https://doi.org/10.5281/zenodo.5500088) with a GC content of 37.4%, a length of 1356 bases, and 23 unique protospacers (S7A Fig). Circ-2 represented 11 small TR sequences (see here: https://doi.org/10.5281/zenodo.5500088) with a GC content of 62.6%, a length of 1872 bases, and nine unique protospacers (S8 Fig). The genome comparisons indicated that circ-1 and circ-2 were nearly complete sequences of circular or tandem genomes (S9 Fig). Next, we searched for a conserved gene among the similar genomes. From circ-1 and its similar sequences, no consistently shared gene was predicted. Circ-2 and its similar sequences shared one coding gene that was 114 codons in length; however, the protein sequence showed no significant hit in the PDB database (an ORFan). Thus, circ-1 and circ-2 seem to be CRISPR-targeted DNA elements without a reliably detected or annotated coding gene.

The predicted targeting host of circ-1 was *Veillonella*, a common gut *Firmicutes*, based on the connection between the CRISPR DR and the associated protospacers. The DR sequence was aligned to the *Veillonella* genomes LR778174.1 and AP022321.1 (S7C Fig). The DR-aligned loci encode Cas9, Cas1, and Cas2. Therefore, the spacers aligned to circ-1 are likely derived from genuine Class 2 CRISPR–Cas systems [34,42] encoded in *Veillonella* genomes. The protospacer-adjacent motif (PAM) was TTTN (S7B Fig), as calculated from the adjacent sequences of protospacers on circ-1. Twelve protospacers on circ-1 were adjacent to this motif (S7A Fig), indicating that the CRISPR–Cas system restricts this DNA element. The GC content of LR778174.1 was 38.8%, which was close to the circ-1 GC content. Moreover, circ-1 was not aligned to any bacterial genomes, including *Veillonella*, indicating that this element is not encoded in a cellular genome. We concluded that circ-1 is likely an extrachromosomal element restricted by the *Veillonella* CRISPR–Cas system.

We could not identify the circ-2 targeting host because the DR sequence yielded no significant hits, and circ-2 itself also had no significant hit. However, the high-GC content of circ-2 indicates that its possible host is *Actinobacteria*.

## Comparison of CRISPR-targeted TR sequences with available viral and plasmid sequences

The TR sequences identified in this study were compared with sequences included in virus and plasmid genome databases, including RefSeq virus [43], RefSeq plasmid, IMG/VR [44,45], and GVD [10] (Fig 3). IMG/VR is the largest database of uncultivated viral genomes. GVD is a database of viral genomes that were discovered from human gut metagenome datasets using VirSorter and VirFinder, both of which rely on protein homology, which complements our approach of discovery of viral genomes. Therefore, we consider that these databases are appropriate to validate our results.

Among the 2554 large TR sequences identified here, we found that 1726 TR sequences (67.6%) were represented in RefSeq virus, IMG/VR, or GVD (Fig 3A) using a threshold of 85%

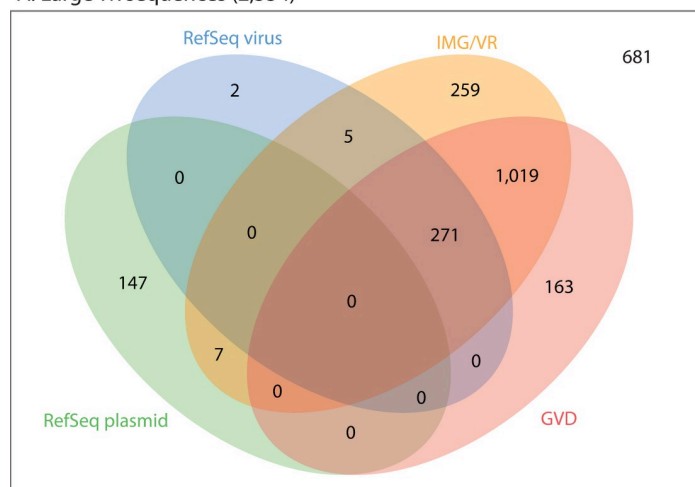
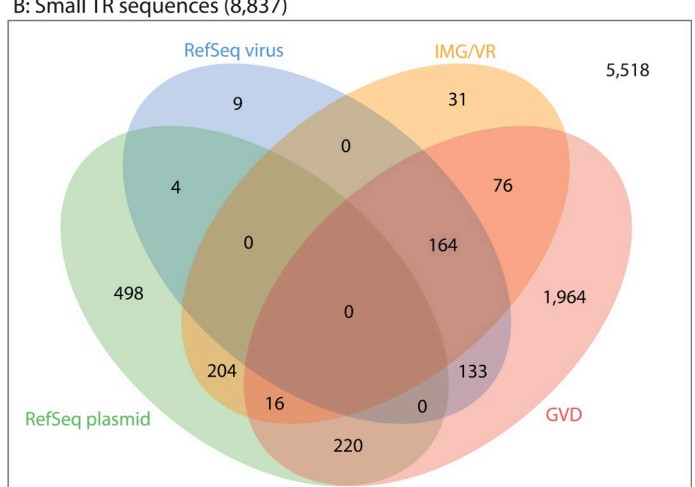

**Fig 3. Venn diagrams of database comparisons for (A) large and (B) small TR sequences.** Each TR sequence was compared to RefSeq virus, RefSeq plasmid, IMG/VR, and GVD using BLASTN. The database hit minimum criteria were set to 85% sequence identity with 75% aligned fraction of the query sequence to a unique subject sequence.

sequence identity with at least a 75% aligned fraction of the query sequence to a unique subject sequence. These sequences included 257 crAssphage genomes ranging in size from 92,182 to 100,327 bases (average, 96,984.8 bases). Members of *Bacteroidales* were predicted as targeting hosts of crAssphages in this study, which is consistent with those hosts reported to propagate crAssphage in previous studies [23,24]. We also found 154 large TR sequences (6%) listed in RefSeq plasmid, seven of which were also listed in the IMG/VR database. Thus, we determined that there is adequate classification between plasmids and viruses for large genomes. Notably, we discovered 11 TR sequences larger than 200 kb, two of which corresponded to recently reported "huge phage" genomes [13] (note in S2 Table); moreover, five of these very large TR sequences encoded HK97 fold capsid proteins. Finally, 681 large TR sequences (26.7%) did not yield database hits using our criteria. We concluded that many of the large TR sequences are already represented in virus or plasmid databases, except for those greater than 200 kb, which were only recently reported.

In contrast with the large TR sequences, most of the small TR sequences were not represented in the databases (Fig 3B). Among the 8837 small TR sequences identified here, 491 (5.6%) were listed in IMG/VR, whereas 2573 (29.1%) were listed in GVD. Only 256 small TR sequences (2.9%) were listed in both IMG/VR and GVD. Finally, 5518 small TR sequences (62.4%) yielded no database hits, indicating that the majority of small elements targeted by CRISPR remain unexplored, even in the intensively studied human gut metagenome. However, IMG/VR filters out genomes shorter than 5 kb, to minimize the rate of false-positive predictions [45], which likely explains the substantially lower representation of small TR sequences in IMG/VR.

Among the 942 RefSeq plasmid-listed sequences, 444 small TR sequences were listed in RefSeq virus, IMG/VR, and/or GVD, suggesting a possible misclassification of viruses and plasmids within these databases. We investigated the database representation of the CRISPR-targeted TR sequences predicted to represent *Microviridae* and *Inoviridae*. Among the 766 putative *Microviridae* genomes from this study, 639 genomes (83.4%) were represented in at least one viral database, and none were listed among RefSeq plasmids. In contrast, among the 56 putative *Inoviridae* genomes, none were listed in either viral or plasmid databases. To

further assess our putative *Inoviridae* genomes, we compared these genomes with recently reported *Inoviridae* genomes discovered using a machine-learning approach [46]. We found that 21 genomes from our study were highly similar to the genomes from the previous study, supporting our prediction that these sequences were indeed *Inoviridae* genomes. Finally, we compared the TR sequences against the Integrative and Conjugative Element (ICEberg) database and obtained nine hits. None of these sequences encoded a detectable capsid protein.

## Comparison with the prediction results of VirSorter

We compared our findings with the prediction results of the VirSorter program. The VirSorter program adopts a homology-based strategy to detect viral genomes; therefore, we considered that this program complemented our strategy. All unique TR sequences were fed into the VirSorter program using default parameters. The program predicted 730 *Microviridae* major capsid-protein-coding TR sequences (95.3%), 916 HK97 fold capsid-protein-coding TR sequences (42.5%), no *Inoviridae* major coat-protein-coding TR sequences, and 109 TR sequences without a detectable capsid predicted to be positively viral (category ≤6). The VirSorter program had a good agreement with the predicted *Microviridae* species identified in our analysis. Conversely, the program predicted about half of HK97 capsid-protein-coding TR sequences and no *Inoviridae* MCP-coding sequences as being positively viral. Notably, among the 257 TR sequences with hits to the crAssphage reference genome, only 10 sequences were predicted to be positively viral (category 3 and 6).

## Classification of *Inoviridae* major coat-protein-encoding TR sequences

Among the discovered capsid/coat-protein-encoding TR sequences, the sequences encoding *Inoviridae* major coat proteins were notably unlisted in the databases; thus, we investigated these sequences regarding whether they contain a novel clade of the viral lineage. Among the 56 *Inoviridae* major coat-protein-encoding TR sequences, 54 encoded the Zonula occludens toxin (Zot). From the Zot-encoding sequences, we selected eight representative genomes, dubbed as Ino-01 to Ino-08 (see here: https://doi.org/10.5281/zenodo.5500088 and note in S1 Table), by clustering Zot amino acid sequences using a 50% sequence similarity threshold; the genus demarcation criteria for the *Inoviridae* family were as proposed by the International Committee on Taxonomy of Viruses (ICTV). The phylogenetic tree of the Zot domains formed a distinct clade, dubbed as Clade-1, which contained six and only discovered genomes (Ino-01 to Ino-06) (Fig 4A). Two other representatives, Ino-07 and Ino-08, were placed in the clades with RefSeq-recorded genomes. The consensus-predicted targeting host class of Clade-1, except Ino-03 and Ino-06, was *Clostridiales*, and the consensus-predicted targeting host phylum of Clade-1, except Ino-03, was *Firmicutes*. Outside Clade-1, the predicted targeting host phylum of Ino-07 was *Proteobacteria*, and the predicted targeting host class of Ino-08 was *Lactobacillales*.

The phylogeny of Zot domains suggests that Clade-1 is the most diversified Zot protein subfamily in the human gut metagenome, and the encoding genomes showed notable diversity as well. The genome lengths of Clade-1 ranged from 5,635 to 9,374 bases, and the GC content ranged from 27% to 41%. Some of the Clade-1 genomes showed nonconventional gene organizations (Fig 4B). The *Zot* genes of Ino-03 and Ino-04 were encoded between the major coat and the probable minor coat gene, which typically is the longest gene and is encoded right after the major coat gene. Ino-06 encoded the tyrosine recombinase after replication initiator gene, suggesting that this virus uses site-specific recombination for viral genome integration into the host chromosome. Accordingly, Ino-06 was partially aligned (43% query coverage with 69.58% sequence identity) to a tRNA locus of *Lachnospira eligens* strain

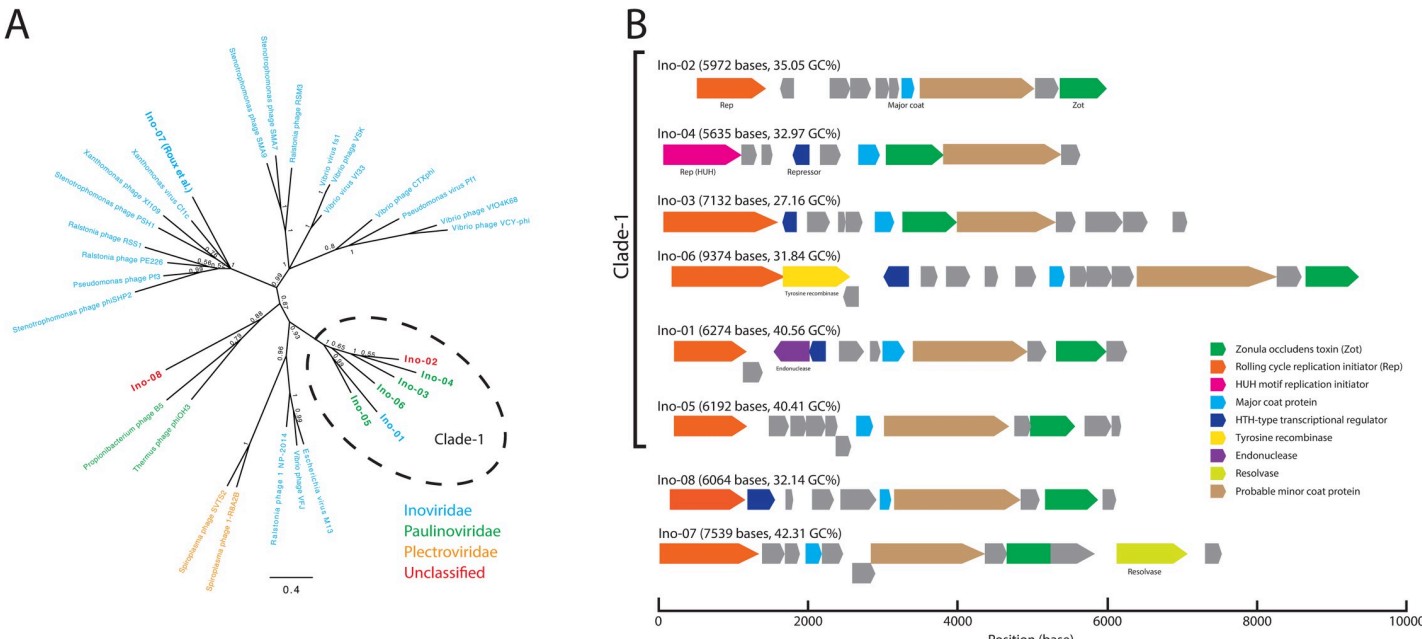

**Fig 4. Classifications and genome organizations of the discovered *Inoviridae* species. (A)** Bayesian phylogeny of Zot domains. Representatives were selected from RefSeq and the *Inoviridae* major coat-protein-encoding TR sequences by clustering Zot amino acid sequences using a 50% identity threshold. Each taxon was colored according to its corresponding family. The families of the discovered genomes (Ino-01 to Ino-08) were predicted using the ICTV-provided taxonomic classification program. A sequence reported in a previous study (Ino-07) is denoted in parenthesis. **(B)** Genome organizations of the discovered *Inoviridae* species. All sequences were phased to align so that the *Rep* genes appear first. The predicted ORFs are colored according to the annotation results.

2789STDY5834875 (accession: NZ_CZBU01000012.1). The aligned region was located at the 3′ end of the *tRNA-lys* gene; this observation demonstrated a previously reported integration mechanism of *Inoviridae* species [47]. Ino-01 encoded an endonuclease in the opposite strand to the essential genes. This endonuclease was partially homologous to homing-endonuclease; thus, we speculated that this virus might use an intron-like mechanism to integrate its genomes into the host chromosome. However, we failed to find a similar sequence to Ino-01 among the RefSeq bacterial genomes. Finally, Ino-04 encoded a HUH-endonuclease domain replication initiator that was non-homologous to the other Rep genes and uncommon in typical *Inoviridae* species [48].

Another notable feature of this study was that all representatives targeted by *Firmicutes* encoded a stand-alone Zot domain, whereas the Ino-07 *Zot* gene was fused to an unknown domain (Fig 4B); a similar structure could be found in other *Proteobacteria*-infecting filamentous phages, such as *M13* and *If1*.

Recently, ICTV updated the taxonomy and definition of the filamentous phage clades. A taxonomy level previously called *Inoviridae* in *ssDNA virus* is now a family of the newly defined Tubulavirales order. This order currently includes three families: *Inoviridae*, *Paulinoviridae*, and *Plectroviridae*. These families were defined based on the analysis of the gene-sharing network of filamentous phage genomes, to represent the enormous HGTs between the closely related species within this order [46,49]. To assign the discovered genomes to the known families based on the gene-sharing method, we used a classification program provided by ICTV [50]. Unexpectedly, the Clade-1 genomes were assigned to two families (Fig 4A); four *Paulinoviridae*, one *Inoviridae*, and one unclassified family. Outside Clade-1, Ino-07 was assigned to the *Inoviridae* family, which was consistent with the other species in the same clade; lastly, Ino-08 was unclassified.

## Gene-content-based hierarchical clustering of CRISPR-targeted TR sequences

We hierarchically clustered TR sequences based on gene content to scrutinize CRISPR-targeted TR sequences in a genomic context. For this analysis, we selected the top 1000 genes that were recurrently observed among large and small TR sequences. The clustering results for large TR sequences (Fig 5A) showed that the majority of genomes, with a variety of gene contents, had already been listed in viral databases. This finding further supports the assertion that these databases contain at least representatives that are similar, at broad taxonomic levels, to the large viruses present in the human gut. Specific gene contents were observed for crAssphages, which Bacteroidetes target exclusively. In addition, RefSeq plasmid hit sequences

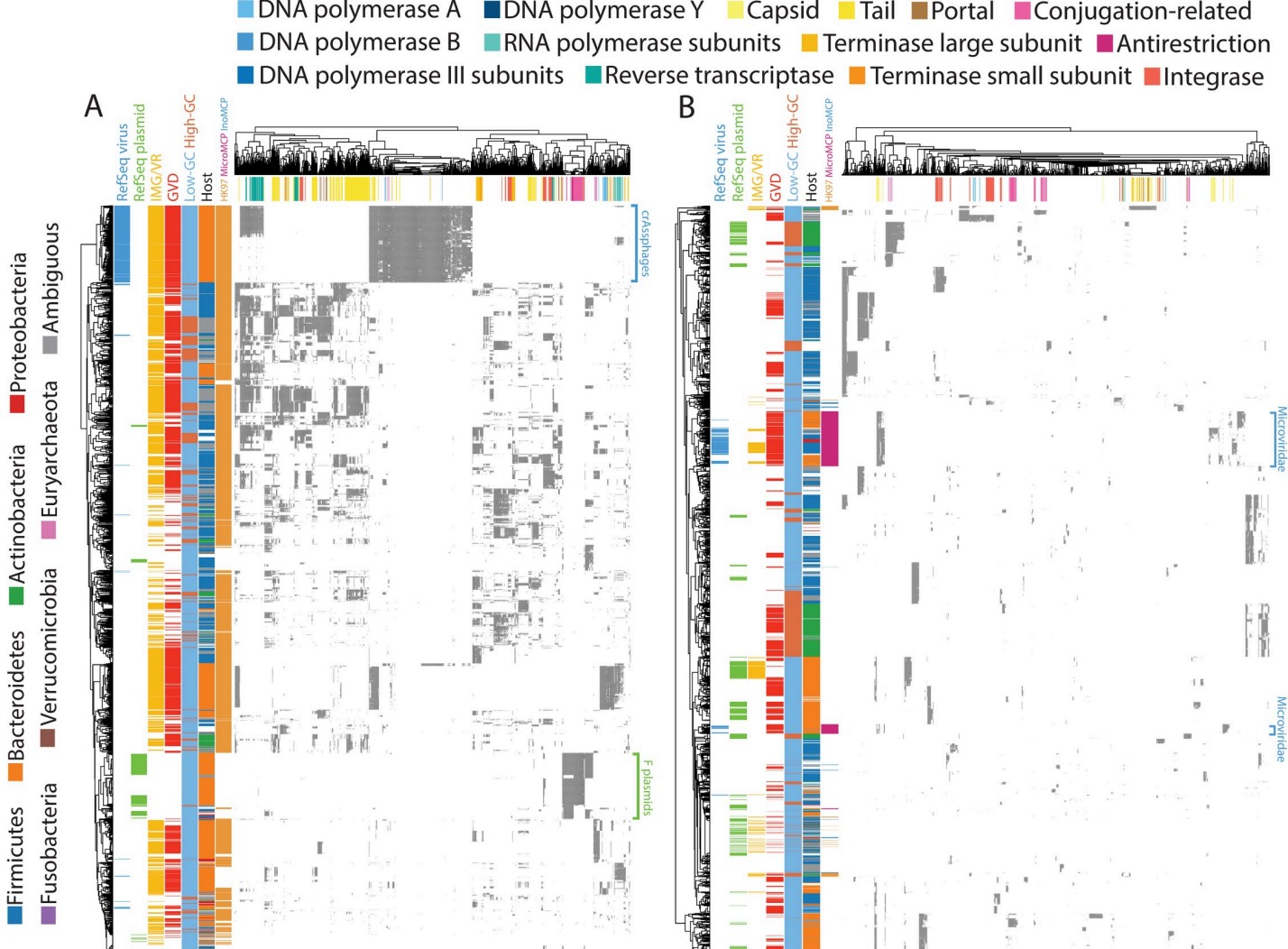

**Fig 5. Hierarchical clustering of (A) large and (B) small TR sequences based on gene content.** Heatmaps representing the gene content of TR sequences, in which each row is a TR sequence and each column is a gene cluster. The gray areas in the heatmap indicate sequences encoding a gene that is homologous to the gene cluster. Note that one gene can be homologous to multiple gene clusters. Sequences are annotated by database containing similar sequences, GC content, host, and capsid genes. Capsid genes are colored differently according to their types, as indicated in the figure; HK97, *Microviridae* major capsid protein (MicroMCP), and *Inoviridae* major coat protein (InoMCP). Gene clusters were annotated by searching corresponding HMMs in the UniRef50 database. Several notable RefSeq-listed clusters are denoted on the right side of the heatmaps.

formed an exclusive cluster with conjugation-related genes. As the conjugation proteins included pili formation and intercellular DNA transfer, these sequences are likely F plasmids. TR sequences were predicted to be targeted by monoderm and diderm hosts clustered separately, indicating little gene flow between them. With the exception of the likely F plasmid sequences, most of these sequences encoded HK97 fold capsids. Combined with the fact that most of the large TR sequences encoded portal proteins, this result further supports the conclusion that the majority of the large TR sequences are *Caudovirales*.

In contrast to the large TR sequences, we found that the small TR sequences remained largely enigmatic when clustered according to gene content (Fig 5B). Few clusters were represented in IMG/VR, and although GVD covers a relatively broader range, representatives were still missing or sparse for some clusters. Several clusters were also listed in both plasmid and viral databases. As many clusters did not encode detectable capsid genes, both clusters representing *Microviridae* evinced capsid genes. Another pattern shown by this analysis was that high-GC-content TR sequences were frequently observed with *Actinobacteria* as the predicted targeting host.

## Remnant CRISPR spacers and contribution of CRISPR-targeted sequences to the identified spacers

Viruses and other MGEs can escape CRISPR targeting by acquiring mutations in protospacer loci [51,52]. Although the corresponding spacers are no longer effective, they can remain in the host genome. To investigate these potential "remnant" CRISPR spacers, we mapped all unique CRISPR spacers to TR sequences and scrambled sequences using various sequence identity thresholds (Fig 6). The scrambled sequences were used to monitor false-positive matches arising by chance (see Materials and Methods). Based on the observation of incremental false-positive matches of spacers to scrambled sequences, we considered that a sequence identity threshold of 84% is adequate for the mapping of some putative remnant spacers, with very few false-positive matches. At an 84% sequence identity threshold, 269,808 and 126,616 spacers were mapped to large and small TR sequences, respectively. Altogether, 20.1% of all unique spacers (396,424 spacers) were mapped to TR sequences. Compared with

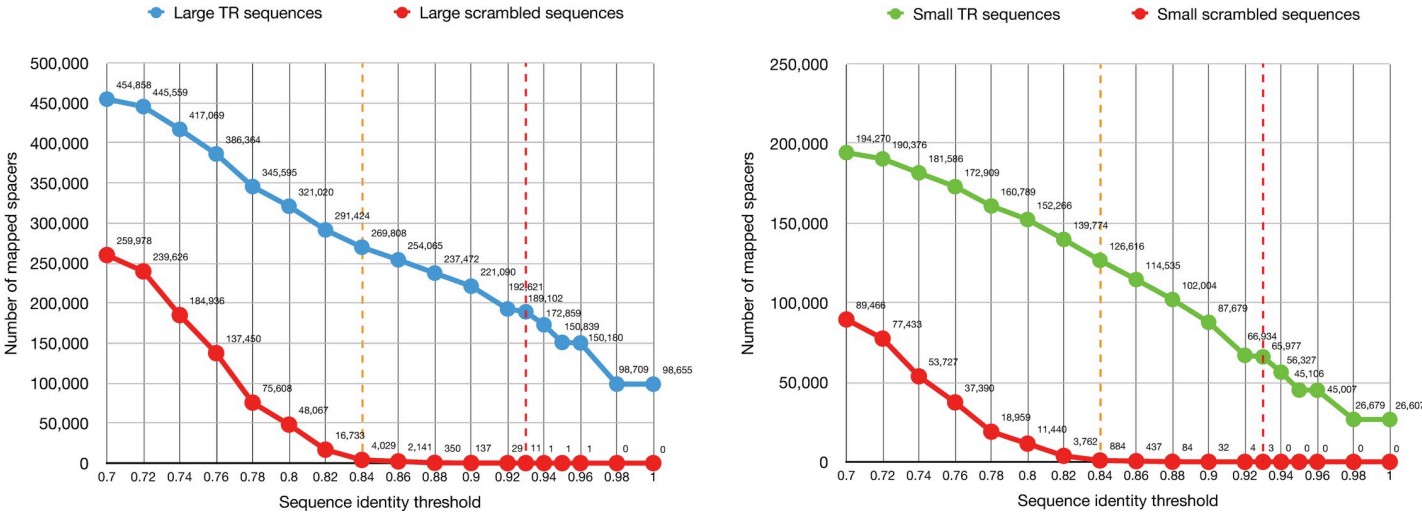

**Fig 6. Number of mapped spacers according to sequence identity threshold.** All unique CRISPR spacers were mapped to large TR sequences, small TR sequences, and scrambled sequences. The relaxed sequence identity thresholds applied initially are denoted as red- and orange-colored dashed lines. The spacer mapping process was identical to the protospacer discovery process (see Materials and Methods).

the identity threshold that was applied initially (93%), 91.9% more spacers were mapped to small TR sequences using the relaxed threshold, whereas only 42.7% more spacers were mapped to large TR sequences. These results suggest that a substantial fraction of CRISPR spacers imperfectly match protospacers within circular MGEs in the human gut, potentially reflecting an "escape mutation" phenomenon. According to percentage, small TR sequences can be explained in this manner better than large TR sequences.

Although we focused on TR sequences because of the high confidence of the genomic completeness, spacers could be derived from incompletely reconstructed or noncircular genomes. To estimate the contribution of the discovered CRISPR-targeted sequences to the identified CRISPR spacers, all unique spacers were mapped to all representative CRISPR-targeted sequences using an 84% sequence identity threshold. Under these conditions, 971,224 spacers (49.3% of all unique spacers) were mapped.

## Discussion

By analyzing the vast size of metagenome sequences, we extracted a large amount of CRISPR-targeted presumably complete sequences of circular genomes. This analysis intended to discover a viral genome that could not be discovered using the conventional homology-based method. Although most of the discovered genomes with detectable capsid genes were previously recognized viral lineages, substantial portions of particularly small TR sequences remained unclassified (Fig 1B). The coding ratio of these unclassified sequences exhibited a broad distribution, and some were exceptionally low; thus, we speculated that these sequences might have unknown genetic features that differ from the conventional protein-coding genes. We selected two small genomes, circ-1 and circ-2, and inferred that one of them was likely extrachromosomal and targeted by the *Veillonella* species CRISPR–Cas system (S7 Fig). These presumable noncoding extrachromosomal DNA elements resemble satellites, which are DNA/RNA elements that replicate with the assistance of the host and/or other MGEs. The one that comes to mind first is the viroid, which is a plant pathogenic circular RNA element that lacks coding genes. However, we cannot infer or relate the discovered genome to known viroids in any biological or evolutionary means based on our current scarce knowledge. Recently reported "satellite plasmids" [53], a plasmid state that lacks autonomous replication genes, also share similarities with these genomes. However, satellite plasmids are an evolutionarily transient state that eventually are lost from the cell population. The genome length and sequence similarity of circ-1 and circ-2 were maintained across various samples, implying that these entire genomic sequences might have unknown functions. Another similar element is represented by circular noncoding RNAs (circRNAs) [54,55]. However, the samples we analyzed in this study did not include RNA sequences, and circRNAs are expressed from cellular genomes, a finding that conflicts with the fact that circ-1 was not aligned to any bacterial genomes, including the presumed host. The function, mobility, and potential pathogenicity of circ-1 and circ-2 remain entirely unknown at this point. Further experimental research and discovery of reassembling DNA/RNA elements are required.

A substantial number of the discovered sequences encoded the HK97 fold capsid, *Microviridae* major capsid, and *Inoviridae* major coat proteins, allowing us to validate that these portions of the discovered genomes were indeed viral. However, in this analysis, vJR capsids were suspiciously absent, despite the fact that vJR capsid-coding viruses, or *Varidnaviria*, are ubiquitous across many environments [56–58]. Currently, we cannot explain why they do not propagate in human gut common bacteria/archaea populations. To date, only two families and nine species belonging to *Varidnaviria* are known to infect bacteria. It is plausible that the lack of a reliable reference genome hampers the detection of vJR capsid genes in the current state.

Considering that nearly half of the detected genes were not annotated with our pipeline, and about 30% of small TR sequences remain unclassified, these remaining sequences may encode vJR capsid genes that cannot be detected based on the currently limited known sequence diversity. A recent application applied a machine-learning approach to this problem and achieved a notable result [59]. The folding-based method could soon complement the sequence-similarity-based method to discover extraordinarily distant homologs.

The targeting host prediction results suggest that approximately 70% of the discovered sequences are targeted by specific host phyla (Fig 2A). Targeting host phyla were ambiguous for 12.5% of the TR sequences; however, most of the targeting host ambiguity observed between *Firmicutes* and *Verrucomicrobia* was suspicious because of the likely horizontal transfer of the CRISPR–Cas system between these species. There is still considerable ambiguity within monoderm phyla. We are uncertain whether these elements infect multiple hosts at present or recently host switched, or whether some genomes became CRISPR-targeted because of abortive infections [60]. TR sequences assigned to a single host taxon were further assessed using a tRNA-based method and cross-study comparison, which yielded good agreement levels above the taxonomic level of order. The human gut metagenome has been intensively sequenced over the past decade, and the database likely captures most of the CRISPR–Cas loci known at present, thus allowing us to confidently predict the targeting hosts of the discovered genomes at higher taxonomic levels (above order). However, in a different environment, i.e., poorly sequenced, the DR-to-RefSeq–based method could lead to significant misinterpretations because of unrecorded HGTs of CRISPR–Cas systems. Therefore, multiple methods should be applied to infer the host of the discovered genomes from such environments. In addition, we state that the spacer-based host prediction method does not directly connect the spacer-aligned sequence to its currently infecting host. A spacer aligned to a sequence is a record of the infection history that occurred in the past, and viruses and MGEs possibly undergo HGTs and switch hosts. Expanding this approach to more diverse samples and observing the evolution of CRISPR loci in particular might allow the inference of the evolutionary history and genetic factors involved in viral/MGE host-switching events.

The disagreement of VirSorter prediction results for HK97-coding sequences could be explained by the high diversity of their gene contents. We found that, among the 257 TR sequences that were similar to the crAssphage reference genome, only 10 sequences were predicted to be positively viral. crAssphages are known to have a variety of gene sets in addition to the essential core gene set, even in a closely related lineage. This might complicate the prediction mechanism of the program, leading to the output of less-confident prediction results. This hypothesis also explains the prediction results of *Microviridae* species. These viruses have small genomes that are densely occupied by few essential genes that are conserved across distant lineages, and such less-diverse gene sets could yield a good prediction agreement. Finally, none of the *Inoviridae* major coat-protein-coding TR sequences, including previously characterized genomes, were predicted to be positively viral. As demonstrated here, none of these *Inoviridae* MCP-coding TR sequences were listed in either viral or plasmid databases. Therefore, this prediction result could be explained by the lack of reference sequences, which precludes the building of a sufficiently sensitive viral gene database for internal use by the program.

The Zot domains of the discovered *Inoviridae* species formed a distinct clade in the phylogenetic tree. The genomes in this clade had notable diversity regarding length and gene component. The predicted families of these genomes were inconsistent, suggesting that further investigation to classify these genomes is required. Filamentous phages, or *Tubulavirales*, are known to undergo intensive HGTs [46,48]. To uncover the whole picture of the complex network of filamentous phage evolution, one might need to build a complete catalog of the

molecular evolution of phage genes, which requires diverse sets of genes collected from various samples. Furthermore, some genes of filamentous phages, including *Rep*, seem to be acquired from non-capsid-protein-coding MGEs, such as plasmids [48]. Applying our method to diverse samples would expand the diversity of virus-MGE shared genes, which could be used to resolve the evolutionary networks of viral genomes.

The results of spacer mapping using a looser criterion suggested that at least one-fifth of the discovered CRISPR spacers originated from TR sequences or their recognizable evolutionary predecessors, whereas about half of the CRISPR spacers originated from our discovered CRISPR-targeted sequences, including both TR and non-TR. The source of nearly half of the CRISPR spacers encoded by residents of the human gut remains unknown, suggesting that additional protospacer reservoirs, whether extinct or simply unsampled, remain uncharacterized.

## Conclusion

We demonstrated that CRISPR spacers can be used to detect viral genomes and other MGEs from metagenome sequences. Using spacers to infer with confidence the sequences that are targeted by CRISPR, we substantially expanded the diversity of MGEs identifiable from the human gut metagenome, which has been a topic of intense investigation for virus discovery. Comparing the sequences predicted by this approach against viral databases showed that our protocol effectively detected viral genomes without requiring similarity to any known viral sequence. Although the majority of large (>20 kb) genomes were predicted as *Caudovirales* with high confidence based on sequence homology, we found that the majority of small (<20 kb) genomes remained unclassified because of a lack of similar genomes in annotated data-bases. Applying this conceptual advancement to additional metagenomic datasets will increase the breadth of the lens through which we can study the diversity of Earth's virome.

## Materials and methods

### Materials

Sequencing data were selected based on NCBI metadata. The filtering parameters used for the query were as follows: layout = PAIRED, platform = ILLUMINA, selection = RANDOM, strategy = WGS, source = METAGENOME, NCBI Taxonomy = 408170 (human gut metagen-ome), and minimum library size = 1 Gb. If a sample contained multiple runs, we selected the run with the most bases, to simplify the analytical pipeline and avoid possible bias to protospa-cer counts from nearly identical metagenomes.

### Database versions and download dates

RefSeq Release 98 was downloaded on January 10, 2020, and IMG/VR Release Jan. 2018 was downloaded on October 21, 2019. GVD was downloaded on March 11, 2020, and UniRef50 was downloaded on December 16, 2019. The PDB database preprocessed by HHsuite was downloaded on September 16, 2020. Metaclust [61] was downloaded on November 10, 2020. The VirSorter database was downloaded on June 7, 2020.

### Metagenome assembly

All downloaded paired FASTQ files were preprocessed based on the guidance provided in BBTools [62] (version 38.73). Adapters, phi X, and human sequences were removed using BBDuk and BBMap. Sequencing errors were corrected using Tadpole. Each preprocessed pair

of FASTQ files was assembled using SPAdes [63] (version 3.12) with the -meta option. Contigs smaller than 1 kb were discarded.

### Detection of CRISPR and spacer extraction

Assembled contigs were scanned with CRISPRDetect [64] (version 2.2) to extract CRISPR DRs, which were deduplicated using CD-HIT-EST [65] (version 4.7) and used to mask the raw reads using BBDuk. We extracted CRISPR spacers from the raw reads to maximize spacer capture from the library. Sequences located between the masked regions within the raw reads were considered CRISPR spacers and were extracted by a simple Python program (available in our source code repository), and then deduplicated.

### Detection of protospacer loci

All DRs were mapped to contigs using BBMap with a 93% minimum sequence identity. The DR mapped positions and their flanking 60 bases were masked as CRISPR loci. Next, the identified spacers were mapped to all CRISPR masked contigs with a 93% minimum sequence identity. Rather than excluding all contigs with CRISPR loci, we exclusively masked CRISPR loci to identify viruses that encode CRISPR systems that are themselves targeted by other CRISPR systems. To increase specificity, we aligned the 5′ and 3′ adjacent regions of spacer-mapped positions. These adjacent sequences were also aligned to the DR sequence associated with the mapped spacer. We discarded loci in which any alignment score divided by the length was higher than 0.5, using the following alignment parameters: match = 1, mismatch = −1, gap = −1, and gap extension = −1. The remaining positions were considered authentic protospacer loci.

### Co-occurrence-based spacer clustering

We clustered spacers in two steps (S2 Fig). First, we clustered protospacer loci located within 50 kb of another protospacer locus. We then clustered spacers based on the co-occurrence of protospacers represented as a graph. In this graph, protospacers are nodes, and the edges represent the co-occurrence of connected protospacers. The weights of edges were the observed counts of co-occurrence of the connected protospacers, as defined in the previous clustering. Graph communities were detected using a Markov clustering algorithm [66] (options: -I 4 -pi 0.4; version 14–137). Clusters with a size smaller than 10 and a global clustering coefficient lower than 0.5 were discarded. Finally, 12,749 clusters comprising 591,189 spacers were derived.

### Extraction of CRISPR-targeted sequences

Contiguous regions of contigs targeted by more than 30% of the members of a spacer cluster were marked as a bed file using BEDTools [67]. To join the fragmented clusters, adjacent regions within 1 kb were concatenated. Marked regions were extracted, and sequences containing assembly gaps were discarded. Finally, both ends of each extracted sequence were compared, to identify TR sequences using a Python program utilizing the Biopython [68] package (available in our source code repository).

### Deduplication of CRISPR-targeted sequences

TR sequences were clustered using PSI-CD-HIT (options: -c 0.95 -aS 0.95 -aL 0.95 -G 1 -g 1 -prog blastn -circle 1). The remaining CRISPR-targeted sequences were clustered twice using linclust [69] (options: —cluster-mode 2 —cov-mode 1 -c 0.9 —min-seq-id 0.95), then clustered again using PSI-CD-HIT (options: -c 0.9 -aS 0.95 -G 1 -g 1 -prog blastn -circle 1).

### Gene prediction and annotation of CRISPR-targeted sequences

Protein-coding genes were predicted from TR sequences using Prodigal (version 2.6.3) with the -p meta option. Each TR sequence was concatenated in silico, and unique predicted genes were selected to recover truncated genes. Predicted protein sequences with partial frags were discarded. The remaining protein sequences were clustered based on a 30% sequence identity threshold using mmseqs [70] (version e1a1c1226ef22ac3d0da8e8f71adb8fd2388a249). HMMs were constructed from each representative sequence using three iterations of jackhmmer [71] (version 3.2.1) against the Metaclust database. The constructed HMMs were then used as queries to search PDB (probability, >80; E-value, <1e−3) using HHsearch (version 3.1.0).

### Assessment of capsid-protein-detection sensitivity and specificity

The sensitivity and specificity of capsid protein detection were assessed using TR sequences similar to the RefSeq-recorded viral and plasmid genomes. Among the 588 TR sequences recorded in RefSeq virus, we detected capsid genes from 577 TR sequences (271 HK97 capsid genes, 306 *Microviridae* MCP genes, and 0 *Inoviridae* MCP genes) (98.13%). Conversely, among the 1096 TR sequences recorded in RefSeq plasmid, we detected capsid genes from 6 TR sequences (6 HK97 capsid genes) (0.55%). Our pipeline successfully detected capsids from reference recorded viral genomes and did not detect them from nonviral genomes with agreeable measures. Accordingly, we conclude that our pipeline has acceptable sensitivity and specificity.

### Targeting host prediction

DRs were mapped to RefSeq bacterial and archaeal genomes using BBMap. A locus with more than three consecutive DR hits within 100 bases was considered an authentic CRISPR locus associated with the mapped DR. DRs mapped to multiple taxa at a given taxonomic level were not taxonomically assigned to that level. The DRs assigned to taxa were used to predict the targeting host. We counted the protospacers linked to taxonomically assigned DRs within a TR sequence. If the count of a given taxon was ≥10 and exhibited higher than 90% exclusiveness, we considered that the corresponding taxon was a targeting host of a given contig. Host predictions were performed for each taxonomic level: species, genus, family, order, class, phylum, and domain.

### tRNA prediction from TR sequences

TR sequences were fed into ARAGORN program [72] using the -gcbact option, which corresponds to the Bacterial/Plant chloroplast genetic code.

### Gene-content-based hierarchical clustering of TR sequences

TR sequences were scanned by HMMs derived from the clustering results of the predicted protein sequences using both TR and non-TR sequences. The scanned result was represented by a binary matrix (score > 60). We selected the top 1000 genes that were recurrently observed within TR sequences. The matrix was hierarchically clustered using ComplexHeatmap [73] (version 2.5.3) and then annotated according to database hits, host, GC content, capsid types, and predicted gene functions. Clustering was performed separately for large and small TR sequences.

### Phylogenetic analysis of *Microviridae* MCP

Representative MicroMCP sequences were selected by clustering all capsid proteins based on an 85% sequence identity threshold throughout the entire length of the protein. Representative and reference protein sequences were aligned using MAFFT [74] (version 7.310) and then

trimmed using trimAl [75] (version 1.4). Aligned sequences were used for Bayesian phylogenetic analysis using MrBayes [76] (version 3.2.7). A mixed substitution model with a uniform prior that converged to Blosum62 (posterior probability = 1.000) was selected. All other priors were set to the default state. Two Markov chain Monte Carlo chains with identical priors were run over ten million generations and sampled every 500 generations. The standard deviation of split frequencies approached zero (0.007837) over the run. The phylogenetic tree was visualized using FigTree [77].

## Phylogenetic analysis of the Zot domain

Representative Zot protein sequences were selected via clustering based on a 50% sequence identity threshold throughout the entire length of the protein. Representative sequences were aligned using MAFFT with the –localpair option. Aligned domains were manually inspected and extracted, then trimmed using trimAl. Aligned domain sequences were used for Bayesian phylogenetic analysis using MrBayes. A mixed substitution model with a uniform prior that converged to Blosum62 (posterior probability = 1.000) was selected. All other priors were set to the default state. Two Markov chain Monte Carlo chains with identical priors were run over twelve million generations and sampled every 500 generations. The standard deviation of split frequencies approached zero (0.002185) over the run. The phylogenetic tree was visualized using FigTree.

## Generation of scrambled sequences

Scrambled sequences are random sequences that were identical to the TR sequences in length. The sequences were generated based on the sampled nucleotide frequencies from the TR sequences using Biopython (available in our source code repository).

## Supporting information

**S1 Fig. Basic workflow used for viral genome detection.** Human gut metagenome libraries were preprocessed to remove adapters, phi X, and human sequences. After correcting sequencing errors, libraries were assembled. Clustered regularly interspaced short palindromic repeats (CRISPR) loci were discovered from the assembled contigs. Consensus direct repeats (DRs) from the discovered CRISPR loci were used to extract spacers, mask the CRISPR loci, and predict the host. All unique CRISPR spacers were mapped to contigs to discover the protospacer loci. Spacers were clustered based on the co-occurrence of the associated protospacers. Sequences targeted by more than 30% of the members of a spacer cluster were extracted and used for further analysis.
(TIF)

**S2 Fig. Spacer clustering based on the co-occurrence of protospacers.** Initially, protospacer loci were clustered based on the distance between them. Within initial clusters, co-occurrences of protospacers were counted and used to construct an undirected graph. The nodes (spacers) in the undirected graph were further clustered using the Markov clustering algorithm. The mean distances between adjacent protospacer loci within clusters were calculated and used to extract CRISPR-targeted sequences. The length and number of protospacers shown here are conceptual and not based on observed data.
(TIF)

**S3 Fig. Number of sequences with a predicted targeting host according to each taxonomic level.**
(TIF)

**S4 Fig. Number of sequences with a predicted CRISPR-targeting host at the taxonomic level of order.**
(TIF)

**S5 Fig. Heterogeneous distribution of TR sequences targeting host ambiguity.** Circle size approximately represents the popularity of the respective host. The bidirectional arrows connect the top two phyla according to host-assigned protospacer counts (i.e., protospacers most often associated with CRISPR DRs are assigned to these two phyla). The numbers on the arrows are counts of the number of TR sequences associated with the connected phyla.
(TIF)

**S6 Fig.** (A) Host prediction comparison between DR-based and tRNA-based methods. (B) Host prediction comparison between MGV and this study.
(TIF)

**S7 Fig. circ-1 protospacers, associated PAM, and Cas genes. (A)** Genomic map of circ-1. The circle represents the circular genome of circ-1. The positions of protospacers are indicated outside the circle. The protospacers with and without PAM were colored magenta and dark gray, respectively. **(B)** PAM of circ-1 protospacers. Both adjacent sequences of protospacer positions up to 10 bases were collected and then aligned, to generate a logo using WebLogo [78]. **(C)** DR-aligned locus of LR778174.1. The cyan bars are the DR-aligned positions. The genes related to the Class 2 Cas system were annotated using colors.
(TIF)

**S8 Fig. circ-2 protospacers and *ORFan* gene.** Genomic map of circ-2. The positions of protospacers and an *ORFan* gene are depicted outside and inside the circle, respectively.
(TIF)

**S9 Fig.** **(A)** circ-1 and **(B)** circ-2 dot plot representations of genome comparisons. The representative and similar genomes were aligned using nucmer [79], then plotted using mummerplot. For circ-1, the 10 most-similar genomes were selected.
(TIF)

**S1 Table. Samples and assembly summary.**
(XLSX)

**S2 Table. CRISPR-targeted TR sequence summary.**
(XLSX)

## Acknowledgments

The majority of analysis has been done on the supercomputer systems at the National Institute of Genetics (NIG) and Okinawa Institute of Science and Technology Graduate University. RS thanks the NIG Research Administrator for scientific discussions.

## Author Contributions

**Conceptualization:** Ryota Sugimoto, Ken Kurokawa, Ituro Inoue.

**Data curation:** Ryota Sugimoto, Phuong Thanh Nguyen, Jumpei Ito, Nicholas F. Parrish, Hiroshi Mori.

**Formal analysis:** Ryota Sugimoto, Luca Nishimura, Phuong Thanh Nguyen.

**Funding acquisition:** Ryota Sugimoto, Ituro Inoue.

**Investigation:** Ryota Sugimoto, Luca Nishimura, Jumpei Ito, Nicholas F. Parrish, Hiroshi Mori, Hirofumi Nakaoka.

**Methodology:** Ryota Sugimoto, Hiroshi Mori, Ituro Inoue.

**Project administration:** Ken Kurokawa, Ituro Inoue.

**Resources:** Phuong Thanh Nguyen.

**Software:** Ryota Sugimoto.

**Supervision:** Jumpei Ito, Nicholas F. Parrish, Hiroshi Mori, Ken Kurokawa, Hirofumi Nakaoka, Ituro Inoue.

**Validation:** Ryota Sugimoto, Luca Nishimura, Phuong Thanh Nguyen, Nicholas F. Parrish, Hiroshi Mori, Ituro Inoue.

**Visualization:** Ryota Sugimoto.

**Writing – original draft:** Ryota Sugimoto.

**Writing – review & editing:** Luca Nishimura, Jumpei Ito, Nicholas F. Parrish, Hiroshi Mori, Ken Kurokawa, Hirofumi Nakaoka, Ituro Inoue.

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
