## [Decision Letter · Decision Letter 0]

10 Feb 2021

Dear Professor Inoue,

Thank you very much for submitting your manuscript "De novo virus inference and host prediction from metagenome using CRISPR spacers" for consideration at PLOS Computational Biology.

As with all papers reviewed by the journal, your manuscript was reviewed by members of the editorial board and by several independent reviewers. In light of the reviews (below this email), we would like to invite the resubmission of a significantly-revised version that takes into account the reviewers' comments.

The reviewers have raised substantial and significant concerns related to the novelty, accuracy, robustness, and validation of the described method and these concerns (as well as others noted in the reports below) should be fully addressed. 

We cannot make any decision about publication until we have seen the revised manuscript and your response to the reviewers' comments. Your revised manuscript is also likely to be sent to reviewers for further evaluation.

Sincerely,

Elhanan Borenstein

Associate Editor

PLOS Computational Biology

Alice McHardy

Deputy Editor

PLOS Computational Biology

Reviewer's Responses to Questions

**Comments to the Authors:**

Reviewer #1: Sugimoto et al present a census of viruses from the human gut by matching CRISPR spacers to sequences carrying protospacers. The approach is straightforward and timely, and the findings are, on the whole, credible and interesting. The multitude of identified small ssDNA viruses, the majority of these previously unknown, is notable.

There are some concerns, though.

1. The cut-off used for protospacer detection, 93% identity, is not as stringent as the authors claim.

This means 2 mismatches for the typical spacer length are allowed, which is not error-proof, so an assessment of the false positive rate is necessary.

2. The authors claim "may entirely novel genomes of unknown taxa", but this claim is misleading because novelty here means failing an extremely stringent threshold, 85% nucleotide identity. It is disingenuous to claim "entire" novelty and actually any novelty as such based on this threshold. Roughly, this result means that a number of viruses could not be included in already know genera. All novelty statements have to be revised to this effect.

3. The "novel" genomes were then searched with capsid protein profiles to determine the fraction of the "novel" sequences that were likely to come from viruses. However, neither the selection of the capsid genes nor the matching criteria are adequately described. Only the cut-off score is indicated which in itself tell nothing about the specificity and sensitivity of the search.

4. There is effectively no attempt to analyze the genomes of "novel" viruses and place them within or outside the current virus classification. To this reviewer, this is a major deficiency. We want to know whether the approach implemented in this paper actually allows one to discover truly novel viruses, those without close evolutionary connections to known families. The current version of the manuscript does not answer this question.

Language: editing by a native English speaker is desirable - the manuscript is not written particularly poorly, overall, but nevertheless, there are quite a few obscure or ungrammatical sentences.

Reviewer #2: The authors present a methodology to detect viral sequences and their microbial hosts based on CRISPR spacers retrieved from assembled contigs, as well as from unassembled read data. They used the suggested approach to perform an analysis of phage-host interactions in gut-associated microbiomes. The authors conclude by performing a phylogenetic analysis of diversity generating retroelement (DGRs) present in crAssphage, an abundant phage in gut microbial communities.

Despite the extensive bioinformatic analyses, I found fundamental flaws and little conceptual novelty in the presented research.

The methodology is based on using assemblies to associate microbial host cell with CRISPR directed repeats (DRs) and then seeking these DRs in raw read data to recruit large spacer databases associated with each host. The last step is using these spacers to detect protospacers, presumably in viruses, that match the spacers, thus linking hosts to viruses. The problem with this linkage of host-DR-spacer-virus approach is that it does not provide a unique mapping from host to virus. Specifically, the host-DR link is problematic, as it has been demonstrated that evolutionary distant bacteria can contain perfectly identical DRs. This occurs due to the high rates of HGT these systems undergo. The consequence of identical DRs in a completely different host is that, unless there is evidence connecting relevant spacer-containing reads to a specific contig, the genuine microbial host cannot be inferred by the suggested process. This is evident even from the results presented by the authors themselves, who state they observe high rates of phages that are reportedly targeted by bacteria from different phyla (“1,418 TR sequences (12.5%) were predicted to be targeted by multiple phyla”). Without evidence to the contrary, the most likely explanation is that these 12.5% of the targets are not infecting bacteria from different phyla (including viruses allegedly infecting both Gram-positive and Gram-negative bacteria), but rather the CRISPR-Cas systems in these hosts cells have identical DRs. This phenomenon is expected to be even more severe in lower taxonomic levels, which more readily exchange genetic material, rendering the host-virus association not informative.

Further, in the title and along the manuscript, the authors suggest that viral sequences can be detected based on CRISPR-matching protospacers. This is misleading, as it is known (and indicated by the authors at some point) that CRISPR spacers do not target viruses exclusively. Retrieved sequences can include plasmid, transposable elements, and even microbial chromosomes. While not abundant, self-targeting spacers have been found and are explained by silencing of CRISPR-Cas systems using different mechanisms, such as anti-CRISPRs or mutations in the flanking repeats.

The target analysis is based on retrieved terminally redundant sequences. The authors state that they have “successfully extracted 11,391 terminally redundant (TR) CRISPR-targeted sequences ranging from 894 to 292,414 bases. These sequences are expected to be complete or near-complete circular genomes”. I find this assumption to be very problematic. The authors do try to substantiate their assumption by searching for phage signature genes in TR sequences and report finding them mostly in large (>20kb) contigs. For short TRs, they mention that they “are unable to strongly conclude whether the remaining small TR sequences are viruses or not at this point.” Still, throughout the manuscript, they treat those TRs as complete genomes or plasmids. Terminally redundant sequences, especially ones as short as 894 bp, could have other origins, which the authors should have considered. Transposons, for example, can be difficult for assembly algorithms to include within contigs and can have terminal repeats, appearing as TR sequences.

A key component in the suggested approach is the detection of spacers in read data. Yet, the authors failed to mention numerous previous studies applying this approach, including tools designed specifically to perform this task, such as CRASS (Skennerton et al., NAR 2013) and MetaCRAST (Moller et al., PeerJ 2017).

Finally, the concluding evolutionary analysis of DGRs in crAssphage seems unrelated to the rest of the manuscript.

Reviewer #3: Sugimoto and colleagues present a de novo virus detection pipeline based on using CRISPR spacers, and apply this approach to over 11,000 human gut metagenomes to detect viral genomes and infer a surprising 70% with a putative host prediction. The authors additionally focus analyses on diversity-generating retroelement (DGR) loci from the widely distributed crAssphage and surmise that there is a common ancestor with human- and baboon-derived sequences. Overall, for a new virus detection pipeline, I would have expected a thorough benchmarking of the methods compared to existing methods (eg, VirSorter or VirFinder) and detailed quality assessment based on the recent MIUViGs standards (Roux et al, Nature Biotech 2019). As written, the pipeline and resultant data are very descriptive and lack a statistically robust assessment of false/true positive recovery rates. Further, it is unclear the level of novelty of this work compared to other approaches to detect uncultivated viruses from metagenomic data. For example, the authors compare the recovery rate of TR sequences >20Kb and nearly all are recovered in existing databases (IMG/VR, RefSeq Plasmid, GVD), with the remaining <20Kb not well represented in databases, but also questionable as to whether they even represent viral sequences. Below the authors should consider several major concerns with the current manuscript, and seek to address these to better validate their pipeline.

Major Comments

In general, the pipeline is fairly well described, but lacks necessary benchmarking. There are a number of unexplained decisions that also need clarification. For example, the 93% sequence identity threshold for masked CRISPR hits is a very permissive threshold to select, and I question what that rationale was. Typically, identical matches or 1-mismatch are allowed given the short sequence length of spacers and to avoid spurious hits. Additional details for why this threshold was selected and a benchmarking of true vs. false positives is needed to ensure the underlying quality of the data is robust.

The crAssphage DGR analysis and inference is poorly described and questionable. In particular, the link between the human gut crAssphage DGRs and the single (?) baboon sequence to infer function/activity is not grounded in much evidence. A more thorough analysis is warranted. Further, any inference of activity should be based on more than a single time point/sequence, and would ideally have longitudinal data for analysis of dynamics/function.

In general, the manuscript would benefit from significant revision to better organize the development/benchmarking of the pipeline and the subsequence analysis of the 11,000 gut metagenomes. For both, the authors should consider focusing on the lines of evidence to support their findings, instead of drawing conclusions based on little evidence. The authors should also thoroughly check for grammar/spelling.

Minor comments:

L. 54: Using the term “parasitic nature” does not accurately describe viral modes of activities, so I would suggest removing this phrase.

L. 59: Metagenomics encompasses more than sequencing host-associated samples, so I encourage the authors to ensure an accurate definition. Further, the authors should also describe that beyond bulk metagenome sequencing from a given environment, laboratory-based approaches to target viral particles and subsequent sequencing (eg, viromics) is another widely used approach to sequence uncultivated viruses.

L.75: A definition of “totally different” is warranted in this context. While reference-based virus detection tools do have limitations, many available tools rely on HMM-based searches which does enable recovery of divergent virus genomic sequences.

**Have all data underlying the figures and results presented in the manuscript been provided?**

Reviewer #1: Yes

Reviewer #2: Yes

Reviewer #3: Yes

PLOS authors have the option to publish the peer review history of their article (what does this mean?). If published, this will include your full peer review and any attached files.

Reviewer #1: No

Reviewer #2: No

Reviewer #3: No
---

## [Decision Letter · Decision Letter 1]

21 Jun 2021

Dear Professor Inoue,

Thank you very much for submitting your manuscript "De novo virus inference and host prediction from metagenome using CRISPR spacers" for consideration at PLOS Computational Biology.

As with all papers reviewed by the journal, your manuscript was reviewed by members of the editorial board and by several independent reviewers. In light of the reviews (below this email), we would like to invite the resubmission of a significantly-revised version that takes into account the reviewers' comments.

We cannot make any decision about publication until we have seen the revised manuscript and your response to the reviewers' comments. Your revised manuscript is also likely to be sent to reviewers for further evaluation.

Sincerely,

Elhanan Borenstein

Associate Editor

PLOS Computational Biology

Alice McHardy

Deputy Editor

PLOS Computational Biology

Reviewer's Responses to Questions

**Comments to the Authors:**

Reviewer #1: The authors have addressed three of my major comments adequately. However, the most important issue, namely, comment #4, on the lack of an appropriate characterization of the identified virus genomes, remains unresolved. I think this type of analysis is a must in a paper on virus discovery.

Reviewer #2: I appreciate the authors’ genuine efforts to address the issues raised in my review, but I still find fundamental problems with the host taxonomy inference, which is key to the study. The authors attempted to address the concern I raised and added an ad-hoc (the results of which support the issues I’ve brought up). However, I find the host inference analysis *conceptually* problematic. The high rates of HGT, acknowledged by the authors, render the host inference ineffective when relying on an assembled reference database such as RefSeq. Due to the frequent HGT events, observing a directed repeat (DR) in a given organism within the reference database does not mean that the DR resides in the same host in the analyzed sample. Thus the host of a virus targeted by a spacer flanked with DRs cannot be reliably determined based on the presence of the DR in a specific host in RefSeq.

The authors try to deal with this to a certain extent by disregarding DRs that appear in RefSeq in multiple hosts. But this assumes that the representation of CRISPRs in various hosts in RefSeq is comprehensive and static. However, in actuality, it is partial and very much dynamic. Thus, the host inference analysis, which is a major part of this manuscript, is highly problematic in its current form.

Following my comment on the matter, the authors referred to plasmids and transposons in different parts of the manuscript. However, the title of the manuscript, as well as the abstract and major parts of the main text, still refer solely to viruses. This is misleading, as more than 59% of the small TR seqeunces and more than 15% of the large TR sequences detected in this study seem to originate from plasmids or similar elements.

**Have the authors made all data and (if applicable) computational code underlying the findings in their manuscript fully available?**

Reviewer #1: Yes

Reviewer #2: Yes

PLOS authors have the option to publish the peer review history of their article (what does this mean?). If published, this will include your full peer review and any attached files.

Reviewer #1: No

Reviewer #2: No
---

## [Decision Letter · Decision Letter 2]

7 Sep 2021

Dear Professor Inoue,

We are pleased to inform you that your manuscript 'Comprehensive discovery of CRISPR-targeted terminally redundant sequences in the human gut metagenome: viruses, plasmids, and more' has been provisionally accepted for publication in PLOS Computational Biology.

Best regards,

Elhanan Borenstein

Associate Editor

PLOS Computational Biology

Alice McHardy

Deputy Editor

PLOS Computational Biology

Reviewer's Responses to Questions

**Comments to the Authors:**

Reviewer #2: The additional analyses and clarifications the authors added to the manuscript text addressed all my concerns.

I appreciate the authors' thorough and elaborate reply.

**Have the authors made all data and (if applicable) computational code underlying the findings in their manuscript fully available?**

Reviewer #2: Yes

PLOS authors have the option to publish the peer review history of their article (what does this mean?). If published, this will include your full peer review and any attached files.

Reviewer #2: No

---

## [Editor Report · Acceptance letter]

27 Sep 2021

PCOMPBIOL-D-20-01978R2 

Comprehensive discovery of CRISPR-targeted terminally redundant sequences in the human gut metagenome: viruses, plasmids, and more

Dear Dr Inoue,

I am pleased to inform you that your manuscript has been formally accepted for publication in PLOS Computational Biology. Your manuscript is now with our production department and you will be notified of the publication date in due course.

With kind regards,

Livia Horvath
